# Evaluation of Ten Alternative Treatments for the Management of Harlequin Bug (*Murgantia histrionica*) on Brassica Crops

**DOI:** 10.3390/plants13121618

**Published:** 2024-06-12

**Authors:** Sarah Clark, Ricardo Bessin, David Gonthier, Jonathan Larson

**Affiliations:** Department of Entomology, S-225 Ag. Sci. Center North, University of Kentucky, Lexington, KY 40546, USA; ric.bessin@uky.edu (R.B.); djgo227@g.uky.edu (D.G.); jonathan.larson@uky.edu (J.L.)

**Keywords:** harlequin bug, sustainable agriculture, Brassica row crops, biopesticides, essential oils, bokashi, pesticide alternatives

## Abstract

Harlequin bug (*Murgantia histrionica*) poses a significant threat to cruciferous vegetable crops, leading to economic losses and challenges in sustainable agriculture. This 2-year field study evaluated the efficacy of exclusion netting and selected biopesticides in controlling harlequin bug populations in a field-grown broccoli crop. Treatments included an untreated control, industry standards Azera and Entrust, and ProtekNet mesh netting. Additionally, three commercial essential oil treatments including Essentria IC-3, Nature-Cide, and Zero Tolerance were tested along with two bokashi fermented composting products BrewKashi and Oriental Herbal Nutrient (OHN). During both the first and second year of the study, none of the commercially produced essential oil products or bokashi products were effective in controlling harlequin bug or preventing leaf scars. Conversely, ProtekNet consistently provided the highest level of protection against harlequin bugs of all stages as well as leaf damage scars; it also provided the largest broccoli head width and highest yield. Entrust showed similar results compared to ProtekNet, both with the control of harlequin bug life stages and with leaf scars. These findings indicate that both ProtekNet and Entrust are effective organic alternatives for managing harlequin bug on broccoli, while the selected essential oil and bokashi products do not appear to be effective.

## 1. Introduction

Sustainable management of agricultural pests is an integral part of ensuring global food security [1]. Insect pests are one of the main challenges we face in maintaining our food systems, as they cause economic losses of fruits, vegetables, and other food commodities worldwide each year. Even with the use of broad-spectrum pesticides and biological control methods, an estimated 20% or more of all agricultural crop production worldwide is still lost to insect pests [2]. These losses contribute to food insecurity, which disproportionately impacts Indigenous, African, and other communities of color [3]. As the world’s population increases, some argue that large-scale farming has become a necessity. However, agricultural intensification is destructive to insect biodiversity; particularly toward insects that directly benefit crops such as pollinators and generalist predators [4]. Additionally, there are concerns that common insect pests continue to develop resistance to conventional insecticides due to their widespread use in the field [5]. There is an urgent need for alternative tactics to help make crop production more sustainable. This study addresses those issues by evaluating alternative treatments to effectively manage a particularly problematic crop pest, the harlequin bug.

Harlequin bug (*Murgantia histrionica*) is an invasive hemipteran pest from Mexico and South America. This pest attacks a variety of cruciferous crops, damaging them by inserting its piercing-sucking mouthparts into plant tissue, injecting digestive enzymes, and sucking out fluids. High populations of harlequin bug can cause wilting, necrosis, and even plant death; especially for young plants [5,6,7,8]. This damage reduces the appearance of leaves at harvest, making plants less marketable. Traditionally, harlequin bug has been difficult to manage due to its mobility and its chemical defenses to deter natural enemies, both of which result in a lack of predators [9]. Indeed, harlequin bug has few natural enemies apart from generalist predators and hymenopterous egg parasitoids, making biological control challenging and largely ineffective [10].

Early management strategies of harlequin bug include hand-picking pests off crops, the destruction of overwintering habitat such as field debris, mulching, mowing, and trap cropping with preferred host plants [7,8,10]. More recently, harlequin bug has been easily controlled with the development of broad-spectrum insecticide groups including organophosphates, carbamates, pyrethroids, and neonicotinoids [7]. However, the growing presence of harlequin bug suggests that the pest could be developing resistance to these conventional pesticides [7,10].

While pesticides play an important role in agricultural food production by protecting crops from pests and diseases, the misuse of pesticides may lead to residues in produce, potentially compromising food safety along with the environment. Chronic low-level pesticide exposures, such as through consumption of commercially produced fruits and vegetables, are harmful to human health and have been linked to depression, neurodegenerative diseases, and birth defects [11]. Higher risk compounds are found in groups such as organophosphates, carbamates, pyrethroids, and organochlorines [12]. Not all pesticides will affect the population in the same way. However, we can mitigate harmful residues by using reduced-risk alternatives such as biopesticides; many of which are certified organic.

Organic certification is a process that certifies an agricultural operation’s compliance with organic production rules, allowing their produce to be labeled as such [13]. Currently, certified organic products are part of a rapidly expanding global marketplace [14]. As our focus shifts toward sustainability, i.e., a reduction in synthetic inputs, reestablishing biodiversity, and regenerative agriculture, many growers are implementing organic practices. However, there are still a wide range of production challenges. In a recent national survey of organic farmers [15], controlling insect pests was among the top five areas of concern. Unfortunately, many of the available treatments meeting organic standards underperform or have been found to be of low efficacy. Thus, we must form a cooperative network of scientists, growers, policy makers, and industry to eliminate these barriers and move forward with more sustainable farming practices [16].

This study is focused on resolving a critical and pressing pest insect problem with few effective organic alternatives. Moreover, it impacts organic growers that raise crops attacked by harlequin bug through a field-verified evaluation of alternative methods to manage harlequin bug in more sustainable ways. This study also explores new and differing treatments including essential oils, soil bacteria, and microbial inputs and evaluates their efficacy in controlling harlequin bug populations. These are potentially important biopesticides that should be used to our advantage. The following bioinsecticides were utilized in this study: neem, pyrethrin, spinosad, selected essential oils, soil bacterium *Chromobacterium subtsugae*, and key active ingredients in bokashi composting including *Lactobacillus casei* along with other fermented microbes derived from roots and bark. Note that this is one of the first studies to investigate the potential pest controlling properties of commercially produced bokashi fermented composting products. Neither the selected essential oil products nor the bokashi fermented composting products were effective in controlling harlequin bug populations in the field. However, it’s clear that further knowledge on the efficacy of organic methods is necessary to support the continued adoption of organic practices by providing improved management strategies and data for harlequin bug control.

## 2. Results

### 2.1. Fall 2022

*Weekly scouting data.* There were only 274 total harlequin bug adults recorded during the 2022 season (Table 1). The numbers of small nymphs, large nymphs, and egg masses were much lower (Table 2, Table 3 and Table 4). Results indicated there were no significant differences between the treatment groups for any of the harlequin bug life stage densities. Mean numbers of adult harlequin bugs were not significantly different among treatments (χ^2^(9) = 11.39, *p* = 0.25). With harlequin bug small nymphs, there was also not a significant difference among means (χ^2^(9) = 13.56, *p* = 0.14). Only 27 harlequin bug small nymphs were observed in the field during the 2022 season. Moreover, only 34 harlequin bug large nymphs were observed in 2022, and no significant differences were detected among treatments (χ^2^(9) = 14.64, *p* = 0.10).

As with adults, small nymphs, and large nymphs, harlequin bug egg mass means showed no significant differences between treatments (χ^2^(9) = 8.51, *p* = 0.48). Finally, there were no significant differences between leaf scar means (χ^2^(9) = 10.62, *p* = 0.30) (Table 5). Results likely lacked significance against the control due to the overall low numbers of harlequin bug in the field during the 2022 season.

*Yield data.* There were significant differences in mean broccoli head width among treatments (F(9,36) = 5.20, *p* < 0.001) (Table 6). Control had the lowest mean head width at 15.80 cm (6.22 in) while BrewKashi had the highest mean head width at 23.01 cm (9.06 in); ProtekNet had the second highest mean head width at 21.59 cm (8.50 in). The Wilcoxon rank sum test indicated that the following treatments produced significantly larger head widths than the control: ProtekNet, Nature-Cide, Zero Tolerance, and BrewKashi. Treatment groups Azera and Entrust were not significantly different from the control.

There were no significant differences found for mean marketable yield across treatments (F(9,36) = 0.31, *p* = 0.97) (Table 7). There were also no significant differences found for mean number of heads among treatments (F(9,36) = 0.63, *p* = 0.77). The total marketable weight for the fall 2022 harvest was 329.05 kg (725.43 lbs), and a grand total of 738 marketable heads were harvested. 

### 2.2. Fall 2023

*Weekly scouting data*. Throughout the fall 2023 season, there were zero harlequin bug adults, large nymphs, or egg masses scouted in units with ProtekNet. Indeed, ProtekNet had the lowest number of these harlequin bug life stages compared to all other treatments in the study. ProtekNet also demonstrated a 98.48% decrease in small nymph population compared to control; and a 97.09% decrease in leaf scars compared to control. When compared to the control, Entrust had similar results showing an 84.21% decrease in the number of harlequin bug adults; an 83.28% decrease in small nymphs; a 90.32% decrease in large nymphs; an 80.00% decrease in egg masses; and a 74.47% decrease in the amount of leaf scars. Entrust was statistically equivalent to ProtekNet in management of all harlequin bug life stages.

Among treatments, significant differences were observed in the means of harlequin bug adults (χ^2^(9) = 31.62, *p* = < 0.001); harlequin bug small nymphs (χ^2^(9) = 21.42, *p* = 0.001); harlequin bug large nymphs (χ^2^(9) = 26.01, *p* = 0.002); harlequin bug egg masses (χ^2^(9) = 25.02, *p* = 0.002); and leaf scars (χ^2^(9) = 30.64, *p* < 0.001) (Table 1, Table 2, Table 3, Table 4 and Table 5). During the fall 2023 season, there was a marked increase in the number of harlequin bug adults, small nymphs, large nymphs, egg masses, and leaf scars. A total of 1584 harlequin bug adults were observed during this portion of the study along with 5403 small nymphs, 1148 large nymphs, 1489 egg masses, and 42,773 leaf scars. The post-hoc Wilcoxon rank sum test revealed that two treatments had significantly lower means of adult harlequin bug adults compared to the control: ProtekNet (*p* < 0.001) and Entrust (*p* < 0.001). When compared to control, this represents a 100% decrease and an 83.85% decrease in the number of harlequin bug adults, respectively. ProtekNet and Entrust had significantly lower means of harlequin bug adults, small nymphs, large nymphs, egg masses, and leaf scars compared to every other treatment (*p* < 0.001).

Entrust demonstrated a high level of control throughout the season, often similar to ProtekNet. With harlequin bug adults, small nymphs egg masses, and leaf scars, Entrust was second only to ProtekNet (*p* < 0.001); and with large nymphs Entrust had significantly lower means than every treatment excluding Azera and ProtekNet (*p* < 0.001). Conversely, OHN showed a 139.50% increase in the number of leaf scars making it significantly different from the control (*p* < 0.001). OHN also had significantly higher means of leaf scars compared to several other treatments including Azera, ProtekNet, and Entrust (*p* < 0.001), as well as Nature-Cide (*p* = 0.006) and Zero Tolerance (*p* = 0.04). Plots with high levels of pest pressure suffered heavy damage while plots treated with Azera, ProtekNet, and Entrust suffered little to no damage (Figure 1).

Although not significantly different from the control, Azera had significantly lower means of harlequin bug life stages and leaf scars compared to several other treatments. With harlequin bug adults, Azera had lower means compared to Grandevo (*p* = 0.04) and Essentria (*p* = 0.004). Similarly, Azera had lower means of harlequin bug small nymphs compared to Grandevo (*p* = 0.003) and Essentria (*p* < 0.001), Nature-Cide (*p* = 0.04), Zero Tolerance (*p* = 0.03), BrewKashi (*p* < 0.001), and OHN (*p* < 0.001). Azera also had lower means of large nymphs than Grandevo (*p* < 0.001), Essentria (*p* < 0.001), Nature-Cide (*p* = 0.004), Zero Tolerance (*p* < 0.001), BrewKashi (*p* < 0.001), and OHN (*p* < 0.001); lower means of egg masses than Essentria (*p* = 0.004) and OHN (*p* = 0.001); and lower means of leaf scars than Grandevo (*p* = 0.008), Essentria (*p* < 0.001), Zero Tolerance (*p* = 0.04), BrewKashi (*p* = 0.003), OHN (*p* < 0.001). Grandevo WDG, Nature-Cide, and Zero Tolerance were not significantly different from the control.

*Yield data.* The null hypothesis was rejected in favor of the alternative hypothesis indicating that mean head width was not the same among treatment groups. This suggests that treatment impacted broccoli head width. Significant differences were found for mean broccoli head width among treatments (F(9,36) = 7.49, *p* < 0.001) (Table 6). BrewKashi had the lowest mean head width at 9.14 cm (3.60 in). Azera had the highest mean head width at 12.93 cm (5.09 in), followed closely by Entrust at 12.83 cm (5.05 in) and ProtekNet at 12.78 cm (5.03 in). Dunnett’s test showed that, compared to the control, units using Azera (*p* < 0.001), Entrust (*p* <0.001), ProtekNet (*p* < 0.001), and Zero Tolerance (*p* = 0.01) had significantly larger head widths. Grandevo WDG (*p* = 0.09), Essentria IC-3 (*p* = 0.08), Nature-Cide (*p* = 0.92), BrewKashi (*p* = 0.99), and OHN (*p* = 1.00) had no significant differences when compared to the control.

Moreover, significant differences were found for mean yield among treatments (F(9,36) = 4.11, *p* = 0.002) (Table 7). Thus, the null hypothesis was rejected in favor of the alternative hypothesis. A post-hoc Tukey HSD test with 95% confidence interval revealed that the ProtekNet units had the highest mean yield at 7.12 kg (15.70 lbs) per treatment block while units treated with Essentria IC-3, OHN, and BrewKashi had the lowest mean yield at 4.06 kg (8.94 lbs) per block, 3.74 kg (8.25 lbs) per block, and 3.23 kg (7.13 lbs) per block, respectively. In fact, yield was 61.27% higher with ProtekNet than with control, while the BrewKashi yield was 26.69% lower than control.

One significant difference was found between mean number of heads among treatments (F(9,36) = 2.47, *p* = 0.03) and that difference was between ProtekNet and BrewKashi (*p* = 0.03). A post-hoc Tukey HSD test indicated that units using ProtekNet had significantly higher numbers of harvestable heads than those treated with BrewKashi. Finally, the total marketable weight for the fall 2023 harvest was 237.51 kg (523.61 lbs), and a grand total of 905 heads were harvested. 

## 3. Discussion

The ten selected treatments in this study have complex interactions within agroecosystems. The essential oil products Essentria IC-3, Nature-Cide, and Zero Tolerance did not provide effective control of harlequin bug in this study. There is literature supporting the use of essential oils to control the related brown marmorated stink bug (*Halyomorpha halys*) in field settings, particularly as a spatial repellent [17]. However, there is little to no information supporting the use of essential oils to control harlequin bug in the field. Regarding essential oils, there is a dearth of research performed in field settings. This is especially true for broccoli crops. Many essential oil studies are performed in lab settings, and while there have been promising results in the lab, there are still many unknowns about how essential oils perform in the field [18].

In addition, the selected essential oil treatments showed very high populations of harlequin bug small nymphs and large nymphs compared to the control. These treatments also had severe leaf scarring due to those increased populations. Essentria IC-3, which had the highest means of all three essential oil products, had over double the number of small nymphs, large nymphs, and leaf scars compared to control. We believe the reason for this could be due to the repellent properties of essential oils. Although these repellent properties do not seem to be effective in repelling harlequin bugs or other major broccoli pests, it seems probable that they are effective in repelling natural enemies. If this is the case, it would allow pest populations to grow rapidly as they are not facing predation.

The bokashi fermented composting products BrewKashi and OHN were also not effective in controlling harlequin bug life stages or leaf scars throughout the study. However, there are opportunities for further research with BrewKashi; especially if used in combination with Bt. During the fall 2022 season, BrewKashi combined with Bt produced the highest marketable yield of any treatment. There could be possible scenarios in which the combined BrewKashi and Bt treatment are useful for increasing yield in broccoli fields, particularly if harlequin bug pest pressure is very low during the season.

Additionally, although BrewKashi and OHN do not currently seem like viable alternatives for large-scale commercial operations at this time, they could be useful for small-scale farmers or CSAs; not necessarily for their pest control properties, but rather for their potential soil-enriching qualities [19,20,21]. There are also opportunities for growers to create their own bokashi-based inoculants on site, either through trial and error or with recipes found through free online resources. Although these two products do not seem to be ideal for pest control, they are thought to increase microbial activity in the soil and theoretically improve yield [20]. If growers develop resources to create inoculants like commercially produced BrewKashi and OHN, they could potentially reduce input expenses [22]. Ongoing research is necessary in determining these possible benefits, both for broccoli and for other types of crops.

Azera, Entrust, and ProtekNet provided the highest level of control of harlequin bug life stages and were also effective in preventing leaf scarring. All three of these products seem to be viable organic alternatives useful for commercial growers for the control of harlequin bug, despite harlequin bug not being specifically listed on either the Azera or Entrust product labels as a target pest. The positive results from Azera are in line with results from other trials [23], while the positive results with Entrust are a new insight into the control of harlequin bug on broccoli. Moreover, although ProtekNet has a high financial cost up front, the nets could be used again for multiple seasons [24], likely a period of between 2–5 years after the initial purchase. If growers have these funds available to them, either personally or through various local funding sources and grants, the use of ProtekNet would likely result in increased quality of broccoli heads and increased yield compared to all other products tested in this study.

Although ProtekNet is very effective at excluding harlequin bugs as well as preventing leaf scarring, it is also effective at excluding many types of natural enemies. A recent study [25] evaluated various types of mesh insect netting for permeability to different beneficial and pest species on apple trees. The netting had five different apertures (square, rectangle, triangle, rhombus, and hexagon) and six different sizes (from 0.4 to 2.8 mm). The study found that more aphid predators and leafroller parasitoids colonized trees covered with larger mesh nets (2.3 mm × 3.4 mm). Additionally, an elongated rectangular-shaped mesh appeared to allow beneficials to access the crop while continuing to provide effective protection against apple pests. Thus, there are netting options available that facilitate access for beneficial insects. Additional research will be necessary to determine which nets strike the ideal balance for broccoli crops.

In conclusion, neither the selected essential oil products nor the bokashi fermented composting products were effective in controlling harlequin bug populations on broccoli or at preventing leaf scarring on plants. However, both ProtekNet and Entrust showed very promising results in controlling this pest. Being a field-based trial, this study is distinct from other studies performed in lab settings. It contains important findings for organic growers who currently have a small arsenal of information to access. While the tested essential oil products did not yield positive results for the control of harlequin bug, these results are still necessary to inform growers; particularly broccoli growers searching for effective alternative methods to control harlequin bug. At this time, it is important to temper expectations with essential oils and their ability to control harlequin bug on broccoli, as none of the selected essential oil products in this study showed strong success in controlling harlequin bug populations. It is also increasingly important to draw distinctions between essential oils tested in lab settings and in field settings. This study provides a foundation for continuing practical knowledge of the efficacy of commercially produced essential oil products in the field against difficult-to-control pests such as the harlequin bug. It also encourages further scientific exploration of bokashi fermented composting products, most notably as microbial soil amendments.

## 4. Materials and Methods 

This study was conducted at the University of Kentucky Horticulture Research Farm in Lexington, Kentucky (GPS 37.972°, 84.535°) within plant hardiness zone 6b [26]. The study had two field trials, the first during the fall of 2022 and the second during fall of 2023. For each trial, scouting data was recorded over a period of 6 weeks. Populations of harlequin bug were monitored, and observations were made regarding each treatment’s efficacy in controlling the pest on the crop. 

Treatment efficacy was estimated using three different types of data generated during weekly scouting and harvest: the first data was the population density of the harlequin bug; the second data was the amount of feeding damage caused by harlequin bug; finally, the third data was the yield and quality. Population counts and feeding damage counts would allow the density of the harlequin bug population to be monitored over the growing season, such that treatment blocks with the highest amount of damage would have the lowest yield. 

### Experimental Design

In both 2022 and 2023, the field site was 15.24 m (50 ft) wide and 91.44 m (300 ft) long (0.40 ha; 0.34 A) and was amended with 45.36 kg (100 lbs) of pre-plant 19-19-19 nitrogen on broadcast before planting. Six rows of black plastic were laid on the length of the field: each 91.44 m (300 ft) long and 76.2 cm (30 in) wide, with 2.13 m (7 ft) centers. A single row of fertigation drip tape was buried down the middle of each black plastic row (Toro^®^ Aqua-Traxx 20.32 cm (8 in) emitter spacing) using a custom drip tape layer.

A randomized complete block design (RCBD) was used to divide the field into 50 experimental units, representing 5 replications of 10 treatments arranged in blocks (Figure 2). Each experimental unit was 9.15 m (30 ft) in length plus 0.91 m (3 ft) between blocks, equaling 10.06 m (33 ft) total (0.0008 ha; 0.0021 A). Treatments began when the economic threshold of 1 harlequin bug per 10 plants was met [27]. Each of the 10 treatments were applied every 7 days on a set schedule with a hand-held CO_2_ spray system and continued for 6 weeks until the broccoli reached maturity. The carrier volume of the CO_2_ sprayer was 2000 mL, or 400 mL per experimental unit. There were three standard hollow cone nozzles (Conejet TX-VS12, TeeJet Technologies, Glendale Heights, IL, USA) on the wand which provided full spray coverage of each experimental unit, spraying at 40 PSI. 

In fall 2022, the F1 hybrid broccoli variety ‘Emerald Crown’ (Rupp Seeds, Wauseon, OH) was seeded in 72-cell trays on July 6. The seedlings were grown inside the greenhouse until July 29, then hardened off outdoors for 17 days before planting in the field. On August 16, broccoli seedlings were planted as a single row with a 45.72 cm (18 in) water wheel transplanter. The water was amended with Sol-U-Gro^®^ starter 12-48-8 (Miller Chemical, Baltimore, MD, USA) at a rate of 2.27 kg (5 lbs) per 378.54 L (100 gal) of water. There were 20 broccoli plants per experimental unit including 2 plants in the front of the row and 2 plants in the back of the row that were excluded from data collection. During planting, ProMix media (Premier Tech Horticulture, Quebec, Canada) was added to the planting holes around the broccoli seedlings to suppress weeds. Teff grass (*Eragrostis tef*) was grown between the rows for further weed suppression. 

Shortly after planting, ProtekNet (nylon mesh spaced 0.35 mm × 0.35 mm, Johnny’s Selected Seeds, Winslow, ME, USA) was installed in the corresponding treatment plots. The netting was 4.27 m × 106.07 m (14 ft × 348 ft) and was cut to fit the experimental unit where it was applied. The ends of each section of netting were burned with a propane torch to prevent fraying. Each unit had 6 galvanized metal hoops spaced approximately 1.52 m (5 ft) apart, anchored in the ground and used as support for the netting. The netting was stretched over the hoops and secured with rock bags around the perimeter. Portions of the net were lifted each week to complete scouting counts, and then secured again.

The first round of insecticide treatments was applied the week of 29 August, and continued weekly to the week of 9 October. All treatments were mixed with 7.5 mL Oroboost adjuvant (Oro Agri, Inc., Fresno, CA, USA) for improved spray coverage and 1 teaspoon DiPel (Valent BioSciences, Osage, IA, USA) for control of caterpillars. The first treatment was a control which was water, 7.5 mL of Oroboost adjuvant, and 1 teaspoon DiPel (Table 8). The same treatment was applied to all other treatments in addition to their experimental harlequin bug controls. 

Weekly scouting was performed during the 6-week treatment period approximately 1–2 days after treatments were applied. Each week, 5 random plants were selected from each of the 50 experimental units with a random number generator. Counts were recorded including the number of harlequin bug eggs, small nymphs, large nymphs, adults, and new fan-shaped feeding scars on each of the 5 randomly selected plants.

The field was harvested over a two-week period beginning the week of October 3 and ending the week of October 10. Just before harvest, widths of the first 10 broccoli heads in each experimental unit were measured to calculate mean head width per treatment, excluding the first 2 heads of the row. I hypothesized that this data would determine whether the amount of insect feeding damage on the plants affected head size, i.e., plants with the most damage would have the smallest head widths. Heads were harvested with a 15.24 cm (6 in) stem based on USDA standards for U.S. No. 1 and U.S. No. 2 grades [28] and combined to determine marketable yield. Heads that did not meet USDA standards were separated and not included in marketable yield calculations. 

There were several challenges during the fall 2022 season including poor seed germination and a significant lack of harlequin bugs in the field. Thus, the methods of the study underwent changes from the first year to the second year. In fall 2023, seed germination improved allowing more plants to be set in the field. The plants were spaced 38.10 cm (15 in) apart rather than 45.72 cm (18 in) apart, increasing the number of plants in each experimental unit from 20 to 24. Additionally, the field was planted approximately two weeks earlier and traps were placed around the field with harlequin bug lures containing the aggregation pheromone murganitol. As a result, the fall 2023 season saw an exponential increase in harlequin bug population.

The same F1 hybrid broccoli, variety ‘Emerald Crown’, (Rupp Seeds, Wauseon, OH, USA) was seeded in 72-cell trays on June 22. The seedlings were grown inside the greenhouse until July 21, then hardened off outdoors for 10 days before planting in the field. On August 1, broccoli seedlings were planted as a single row with a 38.10 cm (15 in) water wheel transplanter. The water was amended with Sol-U-Gro^®^ starter 12-48-8 (Miller Chemical, Baltimore, MD, USA) at a rate of 2.27 kg (5 lbs) per 378.54 (100 gal) of water. There were 24 broccoli plants per experimental unit including 2 plants in the front of the row and 2 plants in the back of the row that were excluded from data collection. During planting, ProMix media was added to the planting holes around the broccoli seedlings to suppress weeds. Teff grass (*Eragrostis tef*) was grown between the rows for further weed suppression. 

ProtekNet was installed in the corresponding treatment blocks in the same manner as the previous year. The first spray treatment was applied the week of August 14, and treatments concluded the week of September 18. All treatments (excluding ProtekNet) were mixed with 16 mL Oroboost adjuvant (Oro Agri, Inc., Fresno, CA, USA). No DiPel was used in 2023. The first treatment was a control which was water and 16 mL of Oroboost adjuvant (Table 8). 

Weekly scouting was performed during the 6-week treatment period approximately 1–2 days after treatments were applied. Each week, 5 random plants were selected from each of the 50 experimental units with a random number generator. Counts were recorded including the number of harlequin bug eggs, small nymphs, large nymphs, adults, and new fan-shaped feeding scars on each of the 5 randomly selected plants.

The field was harvested over a two-week period beginning the week of 18 September and ending the week of 25 September. Just before harvest, widths of the first 10 broccoli heads in each experimental unit were measured to calculate mean head width per treatment, excluding the first 2 heads of the row. In 2023, plants were harvested with 7.62 cm (3 in) stems rather than a 15.24 cm (6 in) stem. This change was made due to concerns about excess weight from stems in the final yield calculations. Also, due to extensive damage in the field caused by heavy pest pressure, many plants produced very small broccoli heads that did not meet USDA marketable standards—particularly those in the BrewKashi and OHN treatment blocks. In the BrewKashi treatment blocks, many of the plants did not produce broccoli heads at all. As a result, mean yield was calculated for the fall 2023 season rather than mean marketable yield.

## 5. Statistical Analysis

The fall 2022 dataset was zero-inflated due to the low population of harlequin bugs in the field, and the fall 2023 dataset did not meet the assumption of normal distribution for the repeated measures ANOVA test. Thus, the nonparametric Friedman test was used as an alternative to the repeated measures ANOVA to detect significant differences between harlequin bug life stages and leaf scars, treatment, and block based on rank totals. Consequently, both the fall 2022 and fall 2023 datasets were individually combined so that there was only one mean per treatment, per block within each year.

The Friedman test was chosen to determine whether treatments impacted the population density of harlequin bug or the amount of leaf-scarring on plants. The test also indicated whether there were any significant differences between treatment blocks. If results were significant, post-hoc pairwise comparisons using the Wilcoxon rank sum test with Bonferroni correction for Type I error were performed.

Yield data met all assumptions for a two-way analysis of variance (ANOVA). This test was performed to assess differences in means between block, treatment, and broccoli head width; block, treatment, and mean marketable yield; and block, treatment, and mean number of marketable heads. Where a significant result was found, a post-hoc Tukey honest significant difference test (HSD) with pairwise comparisons was performed. Dunnett’s Test was also performed to compare significant differences to the control group. R version 4.2.2 was used for all data calculations from this portion of the study. 

## Figures and Tables

**Figure 1 plants-13-01618-f001:**
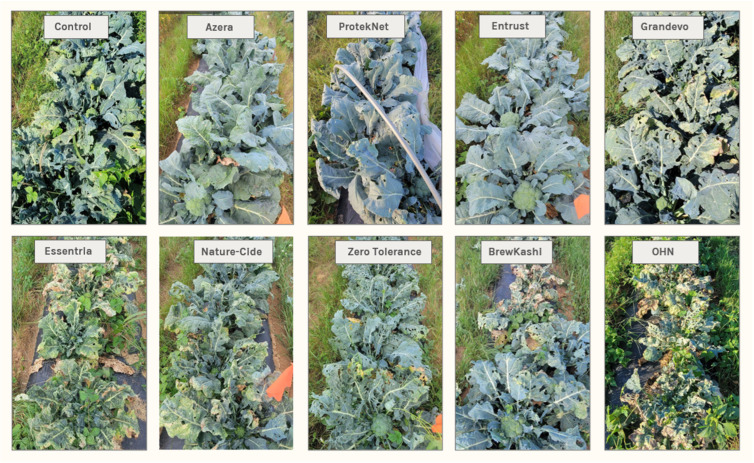
Visual comparison of herbivory during fall 2023. These images show plant damage due to pest pressure in selected plots during the fall 2023 season. Note that plots treated with Azera and Entrust suffered very little damage. Plots with ProtekNet showed minor caterpillar damage, but also suffered very little damage overall. Conversely, plots treated with Essentria, Nature-Cide, Zero Tolerance, BrewKashi, and OHN suffered heavy damage from caterpillars, crucifer flea beetle, and harlequin bug.

**Figure 2 plants-13-01618-f002:**
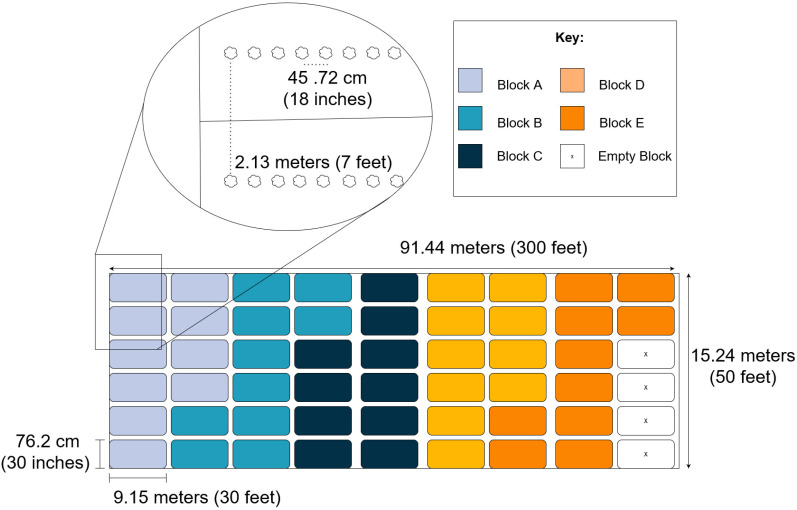
RCBD field layout during fall 2022–2023. This diagram shows the randomized complete block design (RCBD) layout of the field during the fall 2022 and fall 2023 seasons. The length and width of the field are shown in the main diagram, and the broccoli plant spacing is shown in the diagram inset. The five colors represent the five different blocks in the experiment, as shown by the key. Each of the five blocks were composed of 10 experimental units with length and width measurements as shown in the figure. Each experimental unit was randomly assigned a treatment numbered 1–10. The units marked “x” were not part of the experiment and no data was recorded from them.

**Table 1 plants-13-01618-t001:** Harlequin bug adult scouting data.

	Fall 2022		Fall 2023	
Treatment	Season Total	Mean (±SEM)	Season Total	Mean (±SEM)
Control	29	0.19 (0.05)	192	1.28 (0.15) bc
Azera	52	0.35 (0.08)	96	0.64 (0.09) cd
ProtekNet	1	0.01 (0.01)	0	0 d
Entrust	15	0.10 (0.03)	31	0.21 (0.04) d
Grandevo WDG	44	0.30 (0.07)	213	1.42 (0.17) ab
Essentria IC-3	30	0.20 (0.04)	315	2.10 (0.30) a
Nature-Cide	27	0.18 (0.50)	141	0.94 (0.13) bc
Zero Tolerance	19	0.13 (0.13)	155	1.00 (0.15) bc
BrewKashi	32	0.21 (0.05)	214	1.48 (0.19) ab
OHN	25	0.17 (0.04)	227	1.51 (0.15) ab

Total number of harlequin bug adults (*Murgantia histrionica*) per treatment, the mean number per treatment (±SEM), and the mean separation (if applicable). “Mean (±SEM)” shown in the Fall 2022 and Fall 2023 columns represents the mean values per treatment, per 5 plants. Common letters in the same column denote treatment means are not significantly different from one another. Statistics were calculated using a Friedman test comparing treatment groups, followed by a post-hoc Wilcoxon rank sum test.

**Table 2 plants-13-01618-t002:** Harlequin bug small nymph scouting data.

	Fall 2022		Fall 2023	
Treatment	Season Total	Mean (±SEM)	Season Total	Mean (±SEM)
Control	2	0.01 (0.01)	494	3.29 (0.59) bcd
Azera	1	0.07 (0.07)	290	1.93 (0.46) cde
ProtekNet	0	0	8	0.05 (0.04) e
Entrust	0	0	83	0.55 (0.20) de
Grandevo WDG	1	0.07 (0.07)	704	4.69 (0.68) abc
Essentria IC-3	6	0.08 (0.05)	996	6.64 (1.20) a
Nature-Cide	12	0.08 (0.03)	707	4.71 (0.71) abc
Zero Tolerance	0	0	563	3.75 (0.75) abc
BrewKashi	4	0.03 (0.01)	785	5.41 (0.81) ab
OHN	1	0.01 (0.07)	773	5.15 (0.69) ab

Total number of harlequin bug small nymphs per treatment, the mean number per treatment (±SEM), and the mean separation (if applicable). “Mean (±SEM)” shown in the Fall 2022 and Fall 2023 columns represents the mean values per treatment, per 5 plants. Common letters in the same column denote treatment means are not significantly different from one another. Statistics were calculated using a Friedman test comparing treatment groups, followed by a post-hoc Wilcoxon rank sum test.

**Table 3 plants-13-01618-t003:** Harlequin bug large nymph scouting data.

	Fall 2022		Fall 2023	
Treatment	Season Total	Mean (±SEM)	Season Total	Mean (±SEM)
Control	6	0.04 (0.02)	47	0.31 (0.07) bc
Azera	2	0.01 (0.01)	47	0.31 (0.27) bc
ProtekNet	0	0	0	0 c
Entrust	0	0	5	0.03 (0.02) c
Grandevo WDG	1	0.01 (0.01)	185	1.23 (0.33) ab
Essentria IC-3	15	0.10 (0.09)	290	1.93 (0.49) a
Nature-Cide	2	0.01 (0.01)	148	0.99 (0.28) abc
Zero Tolerance	0	0	100	0.67 (0.16) abc
BrewKashi	6	0.04 (0.02)	165	1.14 (0.24) abc
OHN	2	0.01 (0.01)	161	1.07 (0.24) abc

Total number of harlequin bug large nymphs per treatment, the mean number per treatment (±SEM), and the mean separation (if applicable). “Mean (±SEM)” shown in the Fall 2022 and Fall 2023 columns represents the mean values per treatment, per 5 plants. Common letters in the same column denote treatment means are not significantly different from one another. Statistics were calculated using a Friedman test comparing treatment groups, followed by a post-hoc Wilcoxon rank sum test.

**Table 4 plants-13-01618-t004:** Harlequin bug egg masses scouting data.

	Fall 2022		Fall 2023	
Treatment	Season Total	Mean (±SEM)	Season Total	Mean (±SEM)
Control	9	0.06 (0.02)	172	1.15 (0.13) ab
Azera	29	0.20 (0.05)	115	0.77 (0.14) bc
ProtekNet	0	0	0	0 d
Entrust	5	0.03 (0.02)	34	0.23 (0.05) cd
Grandevo WDG	11	0.07 (0.02)	197	1.31 (0.17) ab
Essentria IC-3	16	0.10 (0.04)	203	1.35 (0.16) ab
Nature-Cide	14	0.09 (0.03)	167	1.11 (0.15) ab
Zero Tolerance	13	0.09 (0.01)	168	1.12 (0.14) ab
BrewKashi	9	0.06 (0.03)	179	1.23 (0.15) ab
OHN	4	0.03 (0.01)	254	1.69 (0.20) a

Total number of harlequin bug egg masses per treatment, the mean number per treatment (±SEM), and the mean separation (if applicable). “Mean (±SEM)” shown in the Fall 2022 and Fall 2023 columns represents the mean values per treatment, per 5 plants. Common letters in the same column denote treatment means are not significantly different from one another. Statistics were calculated using a Friedman test comparing treatment groups, followed by a post-hoc Wilcoxon rank sum test.

**Table 5 plants-13-01618-t005:** Harlequin bug leaf scars scouting data.

	Fall 2022		Fall 2023	
Treatment	Season Total	Mean (±SEM)	Season Total	Mean (±SEM)
Control	732	4.88 (0.94)	3190	21.27 (2.20) cde
Azera	1213	8.09 (1.72)	2156	14.37 (2.21) def
ProtekNet	84	0.56 (0.25)	93	0.62 (0.37) f
Entrust	394	2.63 (0.55)	815	5.43 (0.85) f
Grandevo WDG	745	4.97 (1.03)	5352	35.68 (4.29) abc
Essentria IC-3	919	6.13 (1.13)	7876	52.50 (6.29) a
Nature-Cide	1224	8.16 (1.37)	4735	31.66 (4.71) bc
Zero Tolerance	728	4.85 (1.16)	4589	30.59 (3.31) bcd
BrewKashi	923	6.15 (1.18)	6227	42.94 (4.79) ab
OHN	797	5.31 (1.01)	7640	50.93 (4.78) a

Total number of harlequin bug leaf scars per treatment, the mean number per treatment (±SEM), and the mean separation (if applicable). “Mean (±SEM)” shown in the Fall 2022 and Fall 2023 columns represents the mean values per treatment, per 5 plants. Common letters in the same column denote treatment means are not significantly different from one another. Statistics were calculated using a Friedman test comparing treatment groups, followed by a post-hoc Wilcoxon rank sum test.

**Table 6 plants-13-01618-t006:** Broccoli head widths.

	Fall 2022	Fall 2023
Treatment	Mean Head (±SEM in cm)	Mean Head (±SEM in cm)
Control	15.80 (0.93) b	9.63 (0.42) d
Azera	16.80 (1.15) b	12.90 (0.47) a
ProtekNet	21.60 (1.01) a	12.80 (0.48) ab
Entrust	19.50 (0.77) ab	12.80 (0.58) a
Grandevo WDG	19.70 (0.97) ab	11.50 (0.59) abcd
Essentria IC-3	19.80 (1.13) ab	11.50 (0.57) abcd
Nature-Cide	21.40 (0.92) a	10.40 (0.64) bcd
Zero Tolerance	21.30 (0.98) a	12.10 (0.52) abc
BrewKashi	23.00 (0.90) a	9.14 (0.55) d
OHN	20.10 (0.93) ab	9.65 (0.52) cd

For fall 2022 and fall 2023, mean broccoli head widths were compared against treatment groups and blocks using a two-way ANOVA and post-hoc Tukey Test with 95% confidence interval. Common letters in the same column denote treatment means are not significantly different from one another. Head width sizes varied widely in the fall 2022 and fall 2023 seasons due to a marked increase in pest pressure and damage in 2023. Thus, head width during the 2023 season was much smaller on average than head width during the previous season.

**Table 7 plants-13-01618-t007:** Broccoli yield.

	Fall 2022		Fall 2023	
Treatment	Mean Marketable Yield (kg)	Mean Marketable Heads	Mean Yield (kg)	Mean Heads
Control	4.22 (0.71)	9.68 (2.14)	4.41 (0.33) b	18.80 (0.58) ab
Azera	5.15 (0.79)	10.60 (2.87)	6.00 (0.65) ab	18.80 (0.58) ab
ProtekNet	6.85 (0.43)	13.40 (2.66)	7.12 (1.18) a	20.00 (0) a
Entrust	6.58 (0.50)	12.00 (2.56)	5.72 (0.66) ab	18.60 (0.51) ab
Grandevo WDG	7.54 (0.72)	12.80 (3.03)	4.13 (0.24) b	18.00 (0.89) ab
Essentria IC-3	7.65 (1.42)	12.40 (2.45)	4.06 (0.64) b	19.00 (0.32) ab
Nature-Cide	8.09 (1.59)	12.40 (2.41)	4.50 (0.69) ab	17.20 (1.20) ab
Zero Tolerance	8.82 (1.85)	13.00 (2.74)	4.59 (0.68) ab	18.20 (1.32) ab
BrewKashi	9.54 (2.13)	13.00 (2.82)	3.23 (0.34) b	16.00 (1.26) b
OHN	9.35 (2.69)	12.60 (2.49)	3.74 (0.35) b	16.40 (0.87) ab

Yield data for the fall 2022 and fall 2023 field seasons. Note that mean marketable yield and mean marketable heads were calculated in 2022, while mean yield and mean heads were calculated in 2023. Means were calculated using 20 plants per experimental unit. Also listed are the results of the two-way ANOVA performed comparing the mean broccoli yield vs. treatment and block, and mean heads vs. treatment and block. A post-hoc Tukey HSD test with 95% confidence interval was used to calculate mean separation. Common letters in the same column denote treatment means are not significantly different from one another.

**Table 8 plants-13-01618-t008:** Experimental treatments and measurements.

			Fall 2022		Fall 2023	
No.	Name	Active Ingredient	Amount	Water (mL)	Amount	Water (mL)
1	Control			2000		2000
2	Azera	Azadirachtin, Pyrethrins	8 mL	2000	12 mL	2000
3	ProtekNet					
4	Entrust	Spinosad	2 mL	2000	2 mL	2000
5	Grandevo WDG	*Chromobacterium* *subtsugae*	2 tsp	2000	3 tsp	2000
6	Essentria IC-3	rosemary oil, geraniol, peppermint oil	15 mL	2000	31 mL	1900
7	Nature-Cide	cedarwood oil, cinnamon oil	15 mL	2000	27 mL	1900
8	Zero Tolerance	cinnamon oil, clove oil, rosemary oil, thyme oil	125 mL	1900	94 mL	1800
9	BrewKashi	*Lactobacillus casei*	63 mL	1900	78 mL	1900
10	OHN	Fermented angelica root, licorice root, cinnamon bark, garlic, and ginger	2 mL	2000	2 mL	2000

Each of the 10 treatments used during fall 2022 and fall 2023 with their active ingredients. “No.” indicates a treatment’s assigned number in the field. “Amount” indicates the amount of treatment (mL) that was mixed with the corresponding amount of water in the spray tank to treat 5 replicate plots. Note, the treatment and adjuvant concentrations were amended in fall 2023 due to recommendations found through further research.

## Data Availability

Data supporting the reported results can be found at the University of Kentucky’s Institutional Repository under Theses/Dissertations from 2024: “Evaluation of Ten Alternative Treatments for the Management of Harlequin Bug (*Murgantia histrionica*) on Brassica Crops by Sarah Clark” https://uknowledge.uky.edu/entomology_etds/.

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
