# Peer review of "Evaluation of Ten Alternative Treatments for the Management of Harlequin Bug (Murgantia histrionica) on Brassica Crops"

_plants, 2024, doi:10.3390/plants13121618_

Round 1

Reviewer 1 Report

Comments and Suggestions for Authors

The manuscript plants-3026317 entitled “Evaluation of ten alternative treatments for the management of Harlequin bug (Murgantia histrionica) on Brassica crops” tested the efficacy of commercial eco-friendly means of management of Harlequin bug, pest that difficult to control due to high mobility and defenses against natural enemies. Tests were performed in the field with exclusion netting, plant-derived bioinsecticides and bioinsecticides based on microorganisms. The number of individuals at different developmental stages was monitored. ProtekNet and Entrust (spinosad) were the most effective. The number of eggs and adults and number of leaf scars after these treatments was significantly lower than in the untreated group. Mean yield was the highest in ProtekNet group.

Management potential of commercially produced bokashi fermented composting products was studied for the first time.

The manuscript is well written, material and methods are described in detail and results explanations in the Discussion are mostly sound. I only have few suggestions and comments listed below.

 ABSTRACT

-Replace “it also had the largest broccoli head” with “it also provided the largest broccoli head”.

 INTRODUCTION

-Does 35% loss refer to worldwide losses each year? Please, explain in the text.

RESULTS

-“One significant difference was found between mean number of heads among treatments (F(9,36)=2.47, p = 0.03) and that difference was between ProtekNet and BrewKashi (p = 0.03).” I am not sure what you mean. According to small letters in the Table 1.8. ProtekNet also significantly differ from the control, Grandevo WDG and Essentria IC-3.

 DISCUSSION

-Population of adults, small and large nymphs was the highest on Essentria IC-3 (the number is statistically different from the control group). I am not sure that your explanation is correct or sufficient. Within the Introduction you mentioned that Harlequin bug has chemical defenses to deter natural enemies. So, it already deter enemies. Did you find some natural enemies on other treatments? Some essential oils can be attractants for insects which depend on insect species and specific bindings of EO constituents for its chemoreceptors. Therefore, I think that there is a possibility that combination of EOs in Essentria IC-3 has attractant activity for Harlequin bug.

MATERIAL AND METHODS

-Experimental design should be part of Material and Methods. You can put subtitles Experimental design in fall 2022 and Experimental design in fall 2023.

 TABLE 1.1. ““Amount” indicates the concentration of treatment…”. The “Amount” is Amount because it is expressed in mL. You can calculate concentration from the amount of bioinsecticide and amount of water. Please, reformulate this sentence.

 TABLES 1.2.-1.6. In the text for tables you put SE and in the table you put SEM. Use the same abbreviation and put “±” before SE (or SEM). Instead of “Mean per treatment/5 plants” put only “Mean” and explain in caption that it is mean value per treatment/5 plants.

 TABLES 1.7.-1.8. Explain somehow that values in parentheses are SE (or SEM).

 REFERENCES- Article titles in references 1,5,7,8,10,19 should be in sentence case.

Author Response

Note from Sarah Clark: I truly appreciate you taking the time to read through the manuscript. You provided helpful feedback and I found it very thoughtful. Thank you so much for your comments and suggestions.

ABSTRACT
-Replace “it also had the largest broccoli head” with “it also provided the largest broccoli head”.
Revised; highlighted in manuscript.

 INTRODUCTION
-Does 35% loss refer to worldwide losses each year? Please, explain in the text.
Revised, highlighted in manuscript.

RESULTS
-“One significant difference was found between mean number of heads among treatments (F(9,36)=2.47, p = 0.03) and that difference was between ProtekNet and BrewKashi (p = 0.03).” I am not sure what you mean. According to small letters in the Table 1.8. ProtekNet also significantly differ from the control, Grandevo WDG and Essentria IC-3.
The differences you are referring to between ProtekNet, control, Grandevo WDG, and Essentria IC-3 are listed in the "Mean Yield" column rather than the "Mean Heads" column. However, I really appreciate you bringing this to my attention. I was able to find an error in Table 1.8; the mean separations were not provided in the "Mean Heads" column. I added them and they are highlighted in yellow. 

DISCUSSION
-Population of adults, small and large nymphs was the highest on Essentria IC-3 (the number is statistically different from the control group). I am not sure that your explanation is correct or sufficient. Within the Introduction you mentioned that Harlequin bug has chemical defenses to deter natural enemies. So, it already deter enemies. Did you find some natural enemies on other treatments? Some essential oils can be attractants for insects which depend on insect species and specific bindings of EO constituents for its chemoreceptors. Therefore, I think that there is a possibility that combination of EOs in Essentria IC-3 has attractant activity for Harlequin bug.
I did not add the hypothesis that Essentria IC-3 is a possible attractant for harlequin bugs. There is no scientific literature supporting the claim that the essential oil mixtures used in this study attract harlequin bugs. Also, I think mentioning that would require a larger discussion about attractants and repellents that would not necessarily be relevant to this study. I do think it is possible, but I think it is a topic for another study.

MATERIAL AND METHODS
-Experimental design should be part of Material and Methods. You can put subtitles Experimental design in fall 2022 and Experimental design in fall 2023.
Revised, highlighted in manuscript.

TABLE 1.1. ““Amount” indicates the concentration of treatment…”. The “Amount” is Amount because it is expressed in mL. You can calculate concentration from the amount of bioinsecticide and amount of water. Please, reformulate this sentence.
Revised, highlighted in manuscript.

TABLES 1.2.-1.6. In the text for tables you put SE and in the table you put SEM. Use the same abbreviation and put “±” before SE (or SEM). Instead of “Mean per treatment/5 plants” put only “Mean” and explain in caption that it is mean value per treatment/5 plants.
Revised, highlighted in manuscript. Thank you for this clarification!

TABLES 1.7.-1.8. Explain somehow that values in parentheses are SE (or SEM).
Revised, highlighted in manuscript.

REFERENCES- Article titles in references 1,5,7,8,10,19 should be in sentence case.
Revised.

Reviewer 2 Report

Comments and Suggestions for Authors

General comments:

Evaluation of alternative pest management methods is always important for improved IPM strategy and efficient pest control. In the present study, Clark et al. conducted a two years field investigation comparing the control effect of three commercial essential oils, two fermented composting products and insect-proof net for control of Murgantia histrionica on broccoli. This work could provide some evidence for further study and control of this pest.

Overall, the rational presentation of this study has been showed clearly. My main complaints are: 1) Please provide a schematic mapping of the experimental design in the field, as the presented verbal descriptions are not clear; 2) the data showed in the results are not clear, and bar graphs are suggested to show the comparison among different treatments.

Specific suggestions to improve the manuscript are as follows.

Abstract

The Latin name of Harlequin bug should be italic.

Introduction

Line 6-7, the sentence “These losses contribute to food insecurity, which disproportionately impacts Indigenous, African, and other communities of color [3]” is not suitable, should be deleted.

Paragraph 5, line 9-10 “Certain biopesticides also face rigorous regulatory and economic barriers as they continue to develop”. The meaning of the sentence is not clear, please revise it.

Results

Please supply the stats diameters of the ANOVA, including the F value and df.

Discussion

A final conclusion should be proposed at the end of the Discussion section.

Materials and methods

Please provide more information about the block design, such as the interspace between each block and the position settings of each block.

CO2, 2 should be in lowercase.

Comments on the Quality of English Language

Some of the expression should be revised with clearer meaning. 

Author Response

Note from Sarah Clark: Thank you so much for taking the time to read the manuscript and provide suggestions for revisions. We greatly appreciate your input and have responded to your suggestions below.

Overall, the rational presentation of this study has been showed clearly. My main complaints are: 1) Please provide a schematic mapping of the experimental design in the field, as the presented verbal descriptions are not clear;
Thank you for this suggestion. I agree that the experimental design in the field could be made clearer with schematic mapping. It introduces a visual component that makes the field layout easier to understand. We created a new figure (Figure 1.2) and it is now included in the supplemental materials. Caption is highlighted in yellow. 

2) the data showed in the results are not clear, and bar graphs are suggested to show the comparison among different treatments.
This is something that the authors discussed at length, and we ended up deciding to keep the tables rather than replacing them with graphs. We appreciate the visual component that graphs provide, but we feel that these tables are the clearest way we can present the nuanced results of this study. If the reader desires a more visual representation, they can use these tables to create graphs independently.

Abstract
The Latin name of Harlequin bug should be italic.
Revised; hightlighed in yellow in the manuscript.

 Introduction
Line 6-7, the sentence “These losses contribute to food insecurity, which disproportionately impacts Indigenous, African, and other communities of color [3]” is not suitable, should be deleted.
I appreciate and understand this suggestion, however, I think it is important to note this fact simply for social awareness. It is also relevant to the future direction of sustainable agriculture; and additionally, it ties into the next sentence which mentions population increases.

Paragraph 5, line 9-10 “Certain biopesticides also face rigorous regulatory and economic barriers as they continue to develop”. The meaning of the sentence is not clear, please revise it.
Thank you for this suggestion, I agree that sentence was unclear. I ended up just removing the sentence, as I thought the necessary explanation was not really relevant to the study. 

Results
Please supply the stats diameters of the ANOVA, including the F value and df.
I was unclear as to what you meant here; I provided the F-value and degrees of freedom in the text for all ANOVA results. Some of the results in the beginning of the section were from the non-parametric Friedman test and thus had chi-squared values rather than F-values, so those are listed in the text. Please let me know if I misunderstood this suggestion.

Discussion
A final conclusion should be proposed at the end of the Discussion section.
Revised; hightlighed in yellow in the manuscript.

Materials and methods
Please provide more information about the block design, such as the interspace between each block and the position settings of each block.
This sentence was included and highlighted in yellow in the manuscript: "A randomized complete block design (RCB) was used to divide the field into 50 experimental units, representing 5 replications of 10 treatments arranged in blocks (Figure 1.2). Each experimental unit was 9.15 m (30 ft) in length plus 0.91 m (3 ft) between blocks, equaling 10.06 m (33 ft) total (0.0008 ha; 0.0021 A)."

CO2, 2 should be in lowercase.
Revised; hightlighed in yellow in the manuscript.

Round 2

Reviewer 2 Report

Comments and Suggestions for Authors

All my comments have been clearly revised. The manuscript should be accept for publication.